# InCa and InDia: Inline Casing and Diacritization Preprocessing For Robust-to-Noise Tokenization and Interpretability

**Kirill Semenov** [1 2]   **Martin Popel** [2]

## Abstract

We introduce two inline approaches to tokenization preprocessing of casing (InCa) and diacritics (InDia) in the texts. Their main component relies on an automatically created external dictionary that stores information about the most frequent casings or diacritizations of words, and marking only the non-frequent spellings. We show that in a number of noising scenarios, our casing algorithm shows the best performance, and in the cases where it performs on par with the alternative solutions, the intrinsic parameters of the tokenizer trained on our data are more stable. As for inline diacritization, this is the first solution of that type to our knowledge; we show its improvement in robustness against the de-diacritized texts compared to tokenization without preprocessing. We share our preprocessing systems on a public GitHub repository.[1]

## 1. Introduction

The strong point of subword tokenization systems, such as BPE Sennrich et al. (2016) or SentencePiece Kudo & Richardson (2018), is their ability to split any sequence of characters into tokens by falling back to smaller subwords if the character groups are not frequent. However, they are inherently overly sensitive towards variation in character usage. Examples of such variation are various types of casing (capitalization or uppercasing) of the words or omitting the diacritics prescribed by language norms (so-called *de-diacritization*). Tokenizers trained on the general-purpose data usually show poor performance when tokenizing the words which bear similar meaning but are written differently by casing or de-diacritization: since they have not seen enough training examples of different casings or diacritizations, they cannot find the corresponding lines for the upper-cased words and end up over-splitting them into smaller sequences they could find. The illustration of such over-splitting is shown in Table 1.

This variation can be treated as noise and may be deleted beforehand, but in some cases it may also bear linguistic (as in diacritics) or expressive (as in writing the sentences in all caps in the social networks) information. Thus, an ideal solution for handling such a variation would be to preserve the information about casing or diacritization while not damaging the quality of the tokenization. One of the solutions suggested for the casing problem and developed in a line of works is applying preprocessing on the texts before the tokenizer. The recent analysis by Jain et al. (2023) shows that, if applied with a number of tricks (for example, using a single auxiliary token for a sequence of uppercase words) and with data augmentation, it can handle the differently cased inputs well. However, the way the auxiliary symbols are assigned in this paper is questionable, as it allows both treating them as separate tokens and merging them with other words. Moreover, most of the work on the inline casing algorithms focuses either on the downstream performance of the tokenization on NLP tasks or solely on the intrinsic performance of the tokenizers. Bearing in mind that subword tokenization is a relatively new technique and there is still no consensus in how to evaluate the efficiency of the tokenizer itself, this lack of simultaneous analysis of the intrinsic qualities of the tokenizer and the extrinsic performance of the systems which use this tokenizer is a big problem. Finally, to our knowledge, there has been no attempt to apply the inline approach to other orthographical transformations such as diacritics and de-diacritization mentioned above.

In this work, we present InCa and InDia – two systems for preprocessing of texts with respect to casing and diacritization, respectively. Their core characteristic is the usage of an automatically trained dictionary that stores information about the most frequent casing or diacritization of each word in a training corpus, and explicitly marking only the casings

---

[1]Department of Computational Linguistics University of Zurich, Switzerland [2]Institute of Formal and Applied Linguistics, Charles University, Czech Republic. Correspondence to: Kirill Semenov <kirill.semenov@uzh.ch>.

*Proceedings of the ICML 2025 Tokenization Workshop (TokShop)*, Vancouver, Canada. PMLR 267, 2025. Copyright 2025 by the author(s).

[1]https://github.com/Kiryukhasemenov/InFlags

| Variation | Input Phrase | Tokenized Sequence | # Tokens |
|---|---|---|---|
| None | Během výběrů | _Během   _výběr   ů | 3 |
| All-Caps | BĚHEM VÝBĚRŮ | _B  Ě  H  EM  _V  Ý  B  Ě  R  Ů | 10 |
| No Diacritics | Behem vyberu | _Be   hem   _vy   ber   u | 5 |

*Table 1.* An illustration of the tokenization problem with the same phrase transformed either by upper-casing or by deleting the diacritization. The tokenizer trained on the "regular" data (with diacritized and usually not upper-cased words) struggles to split it consistently. The words are overly split, which would influence the processing time and the performance on the downstream NLP task.

| Inline Casing | Sentence | Dictionary |
|---|---|---|
| None | I sold John Baker an iPhone 32 GB and an HTC 64 gb | |
| Basic | T i sold T john T baker an i_ T phone 32 U gb and an U htc 64 gb | |
| InCa | i sold john T baker an iphone 32 gb and an htc 64 L gb | I, John, iPhone, GB, HTC |

*Table 2.* An illustration of our inline casing approach (InCa). The first line is an input sentence containing words with different casing (some spellings, like "gb", are not standardized). Two latter lines show how cased words are substituted with flags (T for title case, U for all caps, L for lower case). Most inline approaches explicitly mark every non-lowercased word (with "iPhone", they also need to handle capitalization inside the word by splitting it). InCa leverages a pre-trained dictionary that stores the most frequent casing of each word (the word "John" is mostly written with title case, and the word "GB" with upper case); therefore it explicitly marks only such casings that are not most frequent (e.g. the surname "Baker" that may also be a common noun). This minimizes the encoded sequence length.

that are less frequent. Our main aim is to create a system that minimizes the lengths of the tokenized sequences, makes the subword vocabularies more transparent (for example, free from doublets differing only by casing) and robust to noise (showing similar tokenization and downstream performance for different types of casing and diacritization of the same text). A draft of this idea was already applied for WMT22 by Popel et al. (2022); in this paper, we present it in more detail, share the code and results of the experiments with this implementation, and extend our approach to the problem of diacritization. A demonstration of this approach compared to other inline casing systems is shown in Table 2.

Our contributions are as follows.

1. In Section 3.1, we introduce a frequency-based inline casing preprocessing algorithm, called InCa (for **In**line **Ca**sing), that is tokenizer-agnostic, fully reversible, and does not require much compute to run.

2. In Section 3.2, we present the first, to our knowledge, **in**line **dia**critization algorithm InDia, also tokenizer-agnostic, fully reversible and easy to run.

3. In Section 4, we test the proposed algorithms on the Czech-Ukrainian machine translation task (MT), evaluating both extrinsic (translation quality) and intrinsic (efficiency of tokenization and tokenizer vocabularies) metrics. For inline casing, we also compare our system with recent inline casing solutions. We show that, compared to the baseline (and to other casing systems), our solutions show stable performance in the general translation scenarios, and for some types of noising (such as full upper-casing) it shows significantly better

downstream performance than any of the compared systems. We also demonstrate that the tokenized sequences become more stable under different noising scenarios, and that the tokenizer vocabularies become more efficient and interpretable in terms of subword uniqueness.

## 2. Related Work

Inline casing approaches were first introduced by Rexline & Robert (2011) as a text compression technique: the upper-cased and title-cased[2] words are substituted with their lower-case correspondences and prepended with the additional symbols that mark the case of the word. In the following, such additional symbols are called **flags** and are denoted in blue. For the MT task, this approach was first implemented by Berard et al. (2019), where it was called "inline casing". There, the flags were applied *after* using BPE to every subword of an upper-cased or title-cased word (which does not sound like an optimal solution). The following papers (e.g. Etchegoyhen & Gete, 2020) returned to the application of casing flags *before* tokenization. Their results show that inline casing is the most efficient case-marking strategy for several MT language pairs, compared to other approaches such as keeping the initial casing, lower-casing, true-casing, recasing, and case factors. The authors suggested that this happened because the inline casing is the strategy that allows "to combine lowercase-based translation benefits with

---

[2]Hereinafter, "upper-cased" refers to a spelling that only consists of capital letters (also known as "all caps"), e.g. "HELLO", and "title-cased" refers to the words starting with a capital letter only, e.g. "Hello World".

case information exploitation". Another study (Shi et al., 2020) compared two variants of placing inline casing flags – either before or after the cased word, and trained the additional neural models for case prediction. They show that all approaches outperform the baseline; specifically, the allocation of flags to the right of a word works better than to the left. However, the inline casing showed lower performance compared to case prediction models.

Recently, two efficient inline casing approaches were published. The first, TokenMonster[3], is a standalone subword tokenization system that includes preprocessing and tokenization modules. In the preprocessing step, two flags (called "capcodes") are assigned; they are responsible for the upper case or the title case, while the input text is transformed to the lower case. In the tokenizer training step, it allows for multiword tokens (by not enforcing token separation by white space), and it uses a variation of the UnigramLM approach (Kudo, 2018), defined as "distillation". As a result, the tokenizer shows text representations in 37.5% fewer tokens at the same vocabulary size compared to GPT-2 or Tiktokenizer[4] used in OpenAI models; the author also trained NanoGPT[5] model on their tokenizer outputs, which showed equal results on several benchmarks such as SQuAD (Rajpurkar et al., 2016) as the pretrained nanoGPT.

Another approach, also integrated into the tokenization system, was presented by the Marian NMT team (Jain et al., 2023). Firstly, their contribution was adding two more flags to two regularly seen upper-case and title-case ones, namely, the beginning and ending of spans of multiple upper-cased words. What is more important, the authors addressed the problem of suboptimality of the encoded sequence lengths made by inline casing systems. Their solution was to distinguish between such words that are often used in the upper or title case from the ones where it happens rarely. In case of frequent usage of cased form (e.g. proper names or abbreviations), the flag is merged with the word and thus is not split: the words "America" and "USA" will be transformed to "Tamerica" and "Uusa", respectively, which allows the tokenizer to merge the flags with the following word; infrequent cases (e.g. regular nouns) are assigned the flags with a white space (thus "Hello" and "HELLO" will be transformed to "T hello" and "U hello"), which would enforce flags as separate tokens. The paper shows increased robustness of the algorithm towards noised casing, as well as only slight changes in encoded length on the noised data compared to the general text. However, it is necessary to note that all algorithms (BPE without preprocessing, classical inline casing, and the proposed algorithms) showed better performance after using the augmented training data.

Other lossless approaches to addressing case variation include: combination of subword embeddings with character representations, augmented by random noise or case toggling (Aguilar et al., 2021); or approaches that are called "case factorization". This term denotes different concepts, for example, handling case information in the same manner as the positional information in the Transformer model – by embeddings that are added pointwise to the word embedding, as in UniCase (Powalski & Stanislawek, 2020); or transformation of strings into 3-dimensional space by variational auto-encoder architecture (Oord et al., 2017), resulting in a set of 3 integers in range [0, 255], as in (Samuel & Øvrelid, 2023).

Regarding the diacritization handling at the tokenizer preprocessing stage, we could not find any research that addresses this topic. The only brief mention and speculation about the impact of diacritics on vocabulary size is made by Alabi et al. (2020), where two low-resource African languages, Twi (which does not use diacritics) and Yoruba (which uses diacritics) are compared by their representations trained in FastText pretrained models (usually of low quality) and on manually curated data. The authors speculate that, despite the fact that Yoruba orthography requires diacritics, there are not many properly diacritized open source data. Otherwise, most NLP solutions either treat the letters with diacritics as the "atomic" characters in the same way as the "base" alphabetic symbols, or strip the text off the diacritics (which seems to happen with consonantal systems and with some large multilingual models such as BERT Devlin et al. 2019).

There is an adjacent body of research related to the restoration of diacritics, both for languages with obligatory diacritics, such as the South Slavic languages (Ljubešić et al., 2016) or Vietnamese (Nga et al., 2019), and for vowel signs for consonantal alphabets, such as Arabic (Shamardan & Hifny, 2023). In most cases, the problem is formulated as an MT task from non-diacritized to diacritized language; thus, solutions such as sequence-to-sequence models or classical statistical MT architectures are applied to it.

Finally, a notable approach similar to inline preprocessing has been introduced for languages with nonconcatenative morphology, such as Hebrew – Splinter (Gazit et al., 2025). The authors suggest separating the root consonants from the consonants and vowels with inflectional meaning, which are orthographically interleaved in a word. The algorithm groups the root characters and the inflectional characters separately, the latter ones in the form of a dictionary, where the key is the position index within a word, and the value is a character.

---

[3] https://github.com/alasdairforsythe/tokenmonster

[4] https://github.com/dqbd/tiktokenizer

[5] https://github.com/karpathy/nanoGPT

## 3. InCa and InDia

Below, we present the algorithms for inline casing and inline diacritization. They share core principles, namely, using a **dictionary** that stores the information about the most frequent casing and diacritization of each word, respectively. Another crucial concept for both methods is a **base**: for both casing and diacritization, a base is a sequence of uncased and undiacritized characters to which all its cased or diacritized versions would correspond. For instance, for the casing, "us" is a base for a lower-cased word form "us", its title-cased counterpart, "Us", and an abbreviation "US". Similarly, for diacritization (using a Czech example), three word forms "zebra" (zebra), "žebra" (rib-GEN.SG), and "žebrá" (beg-PRAES.3.SG) have the same base, "zebra".

### 3.1. InCa - Inline Casing With Dictionary

We introduce **InCa** algorithm, which is the acronym for **In**line **Ca**sing. Its core idea is to collect the counts of each word in the training data about how frequently it occurred in any casing variant (lower, upper, title or any other) and to keep the information about the most frequent version of each word's casing in a dictionary. The system works in three steps:

**1. Training:** For each base (uncased word form) in the training corpus, counts of all possible casings are stored. Then, a **dictionary** is created, which consists of "base": "most frequent casing" items.[6]

**2. Encoding:** Each word[7] in the input text is compared against the dictionary on whether its casing is the most frequent. If it is, the word is transformed into lower case without any flag. If it is not, the word is prepended with a corresponding flag: for the title (T), upper (U) or lower (L) case.[8] There is an explicit lower case flag, contrary to most inline casing systems, since the lower case may not be the most frequent casing of a base (e.g., for proper nouns or abbreviations). The flag and the base are always separated by a single space. For example, the spelling "FRANCE" will be written as "U france" (assuming the most frequent casing is title-cased, "France"). This is done to enforce subword splitting between the casing information and the base at the tokenization stage.

Two additional small tricks are applied to minimize the

---

[6]When storing the dictionary on disk, we can store only the "most frequent casing" variant for each item because the base for can be derived deterministically by lowercasing it.

[7]We split words on alphanumeric/non-alphanumeric boundaries. "Words" without any characters allowing upper or lower case (whitespace, punctuation and other symbols) are kept unchanged by the InCa preprocessing.

[8]There may be words with a casing that is not the most frequent one, nor one of the three standard ones (T, U, L), e.g. "*McDonAld*". Such words are kept unchanged and no flag is prepended.

number of flags even more:

- For sentence-initial positions, we expect the word to be title-cased. Therefore, only the cases where the word is NOT title-cased are marked explicitly with a corresponding L flag.

- For fully upper-cased sentences (lines), we apply a single flag (A for "all upper-case").

**3. Decoding:** The output string that consists of only base spellings and flags is being restored the following way: for each base in the text, we check if it is prepended with an explicit flag and apply the corresponding casing to the word. Otherwise, we check the base against the dictionary and return the most frequent casing from it.

Since the word form distributions in any natural language corpus tend to follow Zipf's law, we can end up with a long dictionary at the training stage, most of the items of which will be the bases seen once or a few times. Thus, we introduced a parameter that sets the **minimal count** of a particular base in the training data to be recorded in the dictionary; otherwise, each word unseen in the dictionary will be explicitly marked if it is not lower-cased.

Compared to other inline casing systems, two objectives of our approach are minimization of the encoded token length and increase in robustness under different casing of the same bases, which happen due to external storage of information about the frequent casing. The only algorithm that attempted to address minimization of token length was the one suggested by Jain et al. (2023); however, their approach allows merging flags with the bases when the words are mostly used in cased forms. This essentially transforms the cased letters into digraphs within the same word, which theoretically should not improve the tokenization length for non-frequent spellings of the words. For example, the word "France" will be tokenized according to this approach as "Tfrance", since it is mostly seen in title case, but for its lower-cased or upper-cased spellings "france" and "FRANCE" the system will assign whole strings to different token sequences. We will show evidence supporting this claim, as well as a comparison with other inline casing approaches in Section 4.

### 3.2. InDia - Inline Diacritization With Dictionary

Inspired by the InCa approach, we leverage it to the problem of diacritization with several modifications. Below, we show the **InDia** method (standing for **In**line **Dia**critization).

**1. Training:** For each base (undiacritized character sequence) in the training corpus, counts of all its possible diacritizations are stored. Then, a **dictionary** is created, which consists of "base": "most frequent diacritization" items.

**2. Encoding:** Each word in the input text is compared

against the dictionary. If it is the most frequent diacritization of the base, it is transformed into a base without a flag. Otherwise, we mark the diacritics that differ from the most frequent diacritization. Since, in many languages, the same diacritic signs can be applied to different characters (or in different positions) in the word, for complete reversibility, we need to keep information about each type of a diacritic sign, as well as its exact character index. This results in inevitable multi-character sequences of the flags. We think of diacritization operations as the dictionary (hereinafter **dict** to distinguish it from high-level dictionaries of InDia), where each key is a character index $id_i$ where a diacritization has to be applied, and $d_i$ is a value, which is an exact diacritization sign. To maximize the compression of the diacritization flag, each flag is stored as a sequence $KV - idx_1 - ID - idx_2 - KV - d_1 - d_2$, where a special symbol $KV$ separates the sequences of keys (in the beginning), and values (in the end), and a special symbol $ID$ separates the indices of the diacritized characters (which are marked by numbers). The reason for making a diacritization flag in the dict form is that such syntax allows for shorter sequences than its main alternative, sequence of diacritization signs $d_1 d_2 ... d_n$ for the whole length of each word. The reason for keeping keys and values on different sides of the flag is our hypothesis that this way, a tokenizer could find frequent patterns for multiple diacritizations independent from the absolute position in a word (and will store them as a single token). A more widespread dict format, $id_1 : d_1, ... id_n : d_n$, does not allow for this since the diacritization signs are separated by character indices.

For example, if "žebra" is the most frequent variant, it will be transformed to "zebra" (the base without any flags), while "žebrá" will be encoded as "KV 4 KV ´ zebra", since only one diacritization of type ´ (acute accent) in character with index 4 differs from the most frequent diacritization. Word form "žebřá" (which does not exist in Czech) would be encoded as "KV 3 ID 4 KV ˇ ´ zebra".

**3. Decoding:** similarly to InCa, at the decoding stage, we look up each base in the dictionary, find its most frequent diacritization, and re-diacritize it according to the dictionary. Then, if the word has an explicit diacritization flag before, we apply all operations mentioned in the flag to the already diacritized version. Notably, the "pivot" diacritization from which we count all differing diacritizations is the most frequent one, not the bare form without diacritizations. Using the most frequent diacritization as a "default" diacritization for each word does not look as evident as for casing; we justify our choice in Appendix A.

To our knowledge, this is the first case of an inline approach to diacritization handling. We also applied two modifications of this approach to see the optimal way of storing the diacritization flags; the comparison will be shown in Section 4.2.

# 4. Case Study: Czech-Ukrainian MT

Experiment Setup

We applied our preprocessing modules to the MT downstream task on the Czech-Ukrainian language pair (both directions). For each observation, we needed to train the full pipeline, which consisted of preprocessing, tokenizer, and MT modules. Therefore, due to compute limitations, our experiments were restricted in the number of languages, preprocessing parameters, and tokenizer choices. Our primary focus was a comparison of different preprocessing solutions. Therefore, the tokenizer and the MT training setups, as well as training and validation data, were fixed for all experiments. The preprocessing details will be explained in two subsections below. The general setup is as follows:

**1. Data:** For training, we used the dataset comprising 8 million sentences that contain all Czech-Ukrainian data from the OPUS corpus (Tiedemann, 2012), WikiMatrix data from the initial publication (Schwenk et al., 2021), and the ELRC EU acts in Ukrainian.[9] For evaluation, the subset of 1012 sentences from Flores 101 dataset was used (Goyal et al., 2022). All data underwent NFKC normalization since it is a default requirement for SentencePiece tokenization (see below) and for treating the diacritization base in InDia.

**2. Tokenizer:** For all setups, SentencePiece (Kudo & Richardson, 2018) implementation of the Unigram LM is used, except for one experiment in casing analysis when TokenMonster was used. All training corpus sentences were used to train the tokenizer. The vocabulary[10] was trained jointly for Czech and Ukrainian, and the vocabulary size was 32,000 tokens.

**3. MT System:** We used the Marian implementation (Junczys-Dowmunt et al., 2018) of the Transformer model Vaswani et al. (2017), specifically, transformer-base model size and 16 epochs for training. All tokenization and MT experiments were run on a single GPU (NVIDIA RTX 3090) for one experiment. The training time typically spanned 22 to 25 hours.

We used a range of extrinsic and intrinsic metrics to evaluate our systems. For extrinsic evaluation, we used BLEU (Papineni et al., 2001), chrF (Popović, 2015) and COMET (Rei et al., 2020); for BLEU and chrF, the SacreBLEU implementation (Post, 2018) was used. For casing experiments, the lowercased versions of BLEU and chrF metrics were

---

[9] The data were taken from elrc-share.eu page.

[10] In what follows, **vocabulary** refers to the set of unique subwords of the tokenizer model, while **dictionary** refers to the auxiliary data structure of InCa and InDia storing most frequent casings and diacritizations.

also used.

Regarding intrinsic evaluation, we selected two metrics based on a comparative analysis of Balhar (2023). Specifically, we use the character per token (CPT) ratio as shown below:

$$CPT(\tau, \pi, C) = \frac{\sum_{s \in C} |s|}{\sum_{s \in C} |\tau(\pi(s))|} \quad (1)$$

Where:

- $\tau$ is a given tokenizer,

- $\pi$ is the preprocessing function such as InCa or InDia (for no preprocessing scenario $\pi(s) = s$),

- $C$ is a given language corpus,

- $s$ is a sentence within the $C$ corpus, $|s|$ is its length in characters and $|\tau(\pi(s))|$ is the length of the encoded sequence in tokens.

This metric aims to estimate the optimality of the encoded text in terms of length. Since we expect a better tokenizer to minimize the length of the encoded sequence, we say that a better tokenizer should have a higher number of characters per token ratio. This metric is a language-independent generalization of metrics such as average sequence length (i.e., average number of tokens per sentence) and word fertility (average number of tokens per word), which are met in other works such as Liang et al. (2023) and Rust et al. (2021).

Another metric used for token distribution estimation is Average Rank (AR). It is defined as:

$$AR(\tau, C) = \sum_{t \in V_\tau} rank(t, \tau(C)) \cdot p(t, C) \quad (2)$$

where

- $\tau$ is a tokenizer function, and $V_\tau$ is its vocabulary,

- $C$ is a given corpus,

- $rank(t, \tau(C))$ is a rank of a token $t$ (position in the list of the unique tokens met in tokenized corpus C ordered by frequency),

- $p(t, C)$ is the frequency of a given token in the corpus.

In other words, the average rank metric is a weighted average of the tokens met in the tokenized corpus, where weights are the frequencies of the tokens in the given corpus. If the distribution is skewed, it will have a long tail of tokens with small probabilities; in this case the bigger frequency weights

will be skewed towards the head of the distribution. The more uniform the distribution (or at least the smaller the tail in favor of the high-frequency tokens), the larger the weighted average. Thus, we expect that the higher average rank of the tokenized text would signify the more optimal usage of the tokens, hence a better tokenizer.

Both intrinsic metrics depend on the validation dataset to which the tokenizer is applied. We are also interested in changes in internal representations of the tokenizer; thus, we evaluate the character per token ratio for the vocabulary items (denoted as $CPT_v$), which is counted as the average number of characters per unique subword in a given tokenizer vocabulary.

Finally, we used another intrinsic metric recently proposed for tokenization evaluation, Rényi Efficiency by Zouhar et al. (2023); however, it showed excessive sensitivity to auxiliary flags used in the tokenized text. We exclude it from the main paper and provide the evidence for problems with this metric application in Appendix C.

### 4.1. Experiments with Casing

Since different methods for inline casing already exist, we are interested in comparing them to our suggested system. Therefore, we compared five modes of preprocessing:

1. **base:** baseline, no inline casing;

2. **inca:** our suggested system;

3. **inca-n:** "naive" version of inca: we analyze how substantial the contribution to the InCa dictionary is; therefore, we exclude the dictionary from the system and explicitly put the flags on every occurrence of the cased word;

4. **marian:** inline casing with diversification by frequency introduced by Marian NMT Jain et al. (2023);

5. **tkm:** TokenMonster preprocessor and tokenizer, that assigns two possible flags and allows for multi-word tokens, thus maximizing the token lengths.

We compare extrinsic performance in both directions of the Czech-Ukrainian translation pair and intrinsic performance in the encoded texts for both languages. We were interested in not only the general MT setup but also three scenarios of case noising: fully upper-cased, fully lower-cased, and with 10% of randomly cased words.

The detailed results with all metrics are presented in Appendix B. For the non-noised scenario, we see that all systems, including the no-preprocessing implementation, go on par, with 21.5-22.0 BLEU score (0.86-0.87 COMET

score) variation for Czech-Ukrainian direction and 22.7-23.3 BLEU score (0.86-0.87 COMET score) interval for Ukrainian-Czech. Similar parity can be seen for lower-cased and 10% randomly cased scenarios. The seeming increase of `inca`, `inca-n` and `marian` approaches for these two noisings (up to 0.5 BLEU point) is not a reliable trend: we estimated the stability of our model by training the `base` and `inca` systems three times, and we obtained the variation of 0.9 BLEU points for both scenarios. Thus, we can see that all preprocessing algorithms, including ours, work on par in general translation task setup.

| Prepro-cessing | BLEU | lc(BLEU) | COMET |
|---|---|---|---|
| base | 1.6 | 1.9 | 0.448 |
| inca | 21.3 | 21.3 | 0.871 |
| inca-n | 20.7 | 20.7 | 0.867 |
| marian | 15.5 | 20.4 | 0.814 |
| tkm | 15.5 | 17.9 | 0.840 |

*Table 3.* Excerpt from extrinsic metrics of the main inline casing algorithms: fully upper-cased noise, Czech-Ukrainian translation pair. Full statistics can be found in Appendix B.

An interesting differentiation occurs in the fully upper-cased scenario. An excerpt of the results for the Czech-Ukrainian direction is shown in Table 3. There, the baseline scores drop down to 1.5-2 points; `tkm` and `marian` systems show moderate performance at around 15-17 BLEU points; and both our systems, `inca` and `inca-n`, almost reach the non-noised quality (20.7-22 BLEU points depending on translation direction). This means that the main difference between the algorithms is that the Marian and TokenMonster casing-trained systems did not output the upper-case flags for the whole sentences (or all words in the sentences). This is supported by qualitative analysis: for instance, the main problem with Marian span marking is that it uses opening and closing flags for upper case sequences, but at the same time, if sequences are interrupted by other cases or non-cased elements, it automatically breaks the uppercasing.

As for intrinsic analysis, we see that for all scenarios except full uppercasing, the baseline, `inca` and `marian` systems perform similarly well, followed by `tkm` and `inca-n`. Under full uppercasing noise, the baseline performance decreases drastically, as does TokenMonster. The general trend in character per token ratio shows that `inca` and `marian` do not significantly differ from the baseline system, which is explained by the fact that the number of the auxiliary flags (and therefore tokens) is intentionally minimized; the slight prevalence of `marian` CPT score (it is stably higher by 0.1-0.2 points than `base` and `inca` that go on par) can be explained by their way of allocating the case markers together with the word itself; therefore the frequent title- or upper-cased words automatically get longer.

Regarding the average rank metric, the best performance is mostly shown by `marian`. However, if we pay attention to consistency of this metric (shown in Figure 1), we will see that the span of AR scores, depending on noise, is larger (especially decreasing under upper case noise), as opposed to `inca`: the total variation of `marian` AR is 120-140 (depending on direction), while for `inca` it is 80-97. Other inline casing systems show even wider spreads, up to 1000 intervals for the baseline scenario. This shows that our system is the most stable under different types of noise.

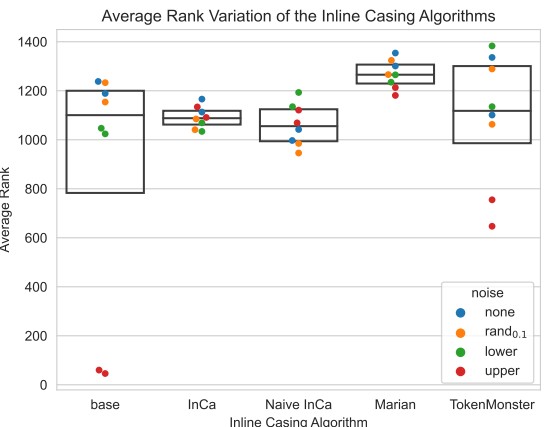

*Figure 1.* Distributions of the Average Rank (AR) metric with respect to different types of noise. Each noise type has two dots denoting texts in Czech and Ukrainian, respectively. Box plots show the median, 25- and 75-percentiles.

The reason for such stability can be seen through the tokenizer vocabulary items. We are interested, first, in whether inline casing helps increase the token length in the tokenizer vocabulary and, second, in how different types of inline casing help release more space for the unique character sequences instead of doubling the tokens that differ only in the casing. Table 4 attempts to estimate that: the $CPT_v$ column provides an answer to the first question, and the "Cased tokens" and "Overlap with Uncased" columns give an estimate for the answer to the second one. We can see that both InCa approaches increase the average unique token length by 0.3 characters. The only approach that beats InCa's is TokenMonster, but this happens because of an uncontrolled orthogonal parameter: allowing the tokens to be multi-word. We can also look at how optimal the inline casing approaches are for saving space for unique lower-case character sequences. Contrary to the no-preprocessing scenario where 19% of unique tokens are cased, and 10% of vocabulary fully corresponds to their lower-cased analogs, we can see that all inline casing algorithms decrease these numbers significantly. However, only the `inca` approaches allow us to decrease these numbers to zero, thus allocat-

ing all possible space released by casing normalization to the new tokens (numbers 4 and 3 in `inca` and `inca-n` columns correspond to the tokens that are flags themselves). Although this may not be directly reflected in the intrinsic metrics above, this is undoubtedly an important feature for the interpretability and predictability of the tokenizer models, as we expect that the casing variation of the tokens would not obscure the variety of the tokens present in the vocabulary.

| Prepro- cessing | $CPT_v$ | Cased Tokens | Overlap with Uncased |
|---|---|---|---|
| base | 6.837 | 6169 | 3508 |
| inca | 7.119 | 4 | 0 |
| inca-n | 7.127 | 3 | 0 |
| marian | 6.554 | 2754 | 1049 |
| tkm | 8.573 | 149 | 92 |

*Table 4.* Tokenizer vocabulary statistics for different preprocessing systems. $CPT_v$ stands for average token length in vocabulary; "Cased tokens" value shows the number of alphabetic tokens that contain a casing flag, and the "Overlap with Uncased" value shows the number of the uncased tokens in the vocabulary that differ from the cased ones (in the "Cased tokens") only by casing or flag prefix.

### 4.1.1. Ablation: Upper-Cased Sentence Flags

In the previous section, we observe that, while performing on par with other systems in default and noisy setups, `inca` leads in a fully upper-cased setup. We hypothesize that this happens due to the specific full-sentence upper-case flag that other systems do not have. To test this, we create a modification of InCa, which only differs in the lack of this flag.

The results in Table 5 indeed demonstrate a significant decrease for the fully upper-cased setup, with a drop by 3 BLEU points. We have already seen such a trend in other algorithms that do not use a special flag for whole sentences - `marian` and `tkm`. Despite that, the InCa without full uppercasing still has 2 BLEU points performance higher compared to the two algorithms mentioned above.

This example tells us that introducing sentence-level flags not only shows intrinsic efficiency in terms of lower encoded

| Prepro- cessing | BLEU | lc(BLEU) | COMET |
|---|---|---|---|
| inca | 21.3 | 21.3 | 0.871 |
| inca-A | 18.0 | 18.3 | 0.850 |

*Table 5.* Extrinsic performance in fully upper-cased scenario, Czech-Ukrainian translation direction. "inca" stands for standard InCa implementation, "inca-A" denotes the ablation without special flags of the full upper-case sentence.

lengths (by putting one flag instead of multiple ones) but also helps downstream performance. This also motivates us to consider introducing other sentence-level flags, for example, for fully title-cased or lower-cased strings.

### 4.1.2. Ablation: Data Augmentation

The authors of the `marian` system, which in some of our setups showed the best extrinsic and intrinsic performance, claimed that they obtained the best performance by combining their inline casing system and leveraging the augmented training data with case variation. We decided to see if our system would benefit from such an option, and to compare it to the baseline system with data augmentation. The augmentation technique was to create a training corpus of the initial data and to add one copy of the training data that is fully upper-cased, one fully lower-cased, and one with 10% of case noise. Thus, we get a training dataset that is four times larger than the initial data; therefore, for comparability, in the augmented setup, we decrease the number of training epochs from 16 to 4.

We compare four modes of preprocessing and MT training pipeline: baseline with and without augmentation and InCa with and without augmentation. The results in the default dataset do not show changes in extrinsic or intrinsic performance; the noticeable changes happen in the fully upper-cased noise scenario (we demonstrate only the results of Czech-Ukrainian direction in Table 6; the other direction shows the same trends). We can see that the extrinsic performance shows a breaking point when using augmentation for the no-preprocessing pipeline, while leveraging InCa does not increase performance.

Does that mean that casing augmentation is a "silver bullet" and we get no improvement from using InCa? To answer this question, we can look at the tokenizer vocabulary statistics. If we compare the average token length in the tokenizers depending on the casing augmentation (see Table 7), we can see that for the non-preprocessing scenario, the casing-augmented tokens became almost 0.5 characters shorter. At the same time, there is no such drop in tokenizers trained after the InCa application. Moreover, if we look at the details of the tokenizer vocabularies, we will see that for the case-augmented no-preprocessing tokenizer, 38% of unique tokens are not lower-cased, and 33% have their full lower-cased analogs in the vocabulary. This demonstrates non-optimal allocation of the vocabulary, contrary to all InCa tokenizers.

### 4.2. Experiments with Diacritization

We conducted several experiments to test our suggestion on inline diacritization. In our language pair, only the Czech language is heavily diacritized (16 letters out of 42 have diacritics); thus, we only apply InDia to the Czech texts. Firstly,

| Augmen-tation | Prepro-cessing | BLEU | chrF | COMET |
|---|---|---|---|---|
| - | base | 1.6 | 22.5 | 0.448 |
| - | InCa | 21.3 | 51.3 | 0.871 |
| + | base | 21.6 | 51.4 | 0.874 |
| + | InCa | 22.2 | 52.2 | 0.877 |

*Table 6.* Extrinsic performance for systems with and without casing augmentation (shown in "Augmentation" column), Czech-Ukrainian direction, fully uppercased noise.

| Augmen-tation | $CPT_v$ | Prepro-cessing | Cased Tokens | Overlap with Uncased |
|---|---|---|---|---|
| - | base | 6.837 | 6169 | 3508 |
| - | InCa | 7.119 | 4 | 0 |
| + | base | 6.495 | 12270 | 10771 |
| + | InCa | 7.205 | 4 | 0 |

*Table 7.* Tokenizer vocabulary statistics for systems with and without casing augmentation. The metrics are described in Table 4.

| Prepro-cessing | BLEU | | |
|---|---|---|---|
| | no noise | fully de-diacritized | 20% de-diacritized |
| base | 21.6 | 9.2 | 18.6 |
| InDia | 21.7 | 17.9 | 21.1 |
| InDia-w | 21.7 | 18.8 | 21.1 |
| InDia-n | 21.0 | 18.4 | 20.5 |

*Table 8.* BLEU scores for different diacritization metrics (by row) under different noise conditions (by column), Czech-Ukrainian translation direction.

we compared the general MT setups for both directions to see if our system showed the same downstream results as the baseline with no preprocessing. It indeed showed consistent performance compared to no preprocessing scenario: InDia shows 21.7 BLEU for Czech-Ukrainian (against 21.6 in the baseline) and 22.8 for Ukrainian-Czech (against 22.7 in the baseline). It is especially notable for the Ukrainian-Czech translation direction, as it shows both the ability of the MT system to learn the token sequences which contain flags, and the InDia decoder allows one to restore the diacritics in the resulting files correctly. The qualitative analysis of the generated diacritization flags for Ukrainian-Czech translation direction shows that, out of 19,760 words in the target text (detokenized after output), there are 642 char-InDia flags, and only 8 of them show hallucinations (in either wrong character index or impossible diacritic-sign combination). Thus, we can reliably use the inline diacritization methods on the output side.

As with inline casing, we are interested in the performance of our system in different noise scenarios. A frequent practice in the Czech online speech is the complete or partial omission of the diacritics in the text. Therefore, we chose two noise scenarios to approximate that: a complete omission of diacritization and omission of diacritization in 20% of words.

Since we are unaware of analogous solutions to diacritization handling, we compare our algorithm with a baseline and two InDia modifications. First is **InDia-n**, a "naive" version of InDia (analogous to inca-n): we do not store information about the frequencies of the diacritizations, thus for every diacritized word in the input text we decompose

it explicitly into the base and the flag consisting of all diacritized characters. Another one is **InDia-w** ("w" stands for "word-level"): there, we use the same frequency-based approach to diacritization as in InDia, but we choose a simpler system of flag notation: we sort all diacritizations of the same base by frequency, and for all diacritizations differing from the most frequent we mention the index of their rank. This makes the flag system shorter (similar to InCa, where each flag is a single character). However, the flags lose their "semantics" (for different bases, the "second frequent rank" may mean different diacritization). The examples of different diacritization systems are shown in Appendix D.

The comparison of different techniques is shown in Table 8. We see that all InDia approaches handle the task significantly better, doubling the quality of the fully de-diacritized text and yielding 3 BLEU points in the 20% de-diacritized text. It is notable, though, that for the fully de-diacritized scenario, the performance of basic InDia is stably lower than of its modifications. Since it lies within 1 BLEU point span, this may be a matter of stability of the NMT training; however, this may be a consequence of how the de-diacritization is marked in the main approach. Specifically, for basic InDia, a specific character-level operation prescribes the deletion of a diacritic (if the most frequent spelling is diacritized). Thus, in the fully de-diacritized scenario, every word whose most popular spelling is diacritized, is prepended with a flag that cancels diacritics for each character. Thus, the encoded length of sequences becomes longer and less informative, affecting the translation quality. At the same time, "naive" InDia does not use any flags for non-diacritized words, and InDia-w uses at most a single-character flag. This is supported by intrinsic metrics in the fully de-diacritized noise scenario. If we look at the character per token ratio, for InDia-n, the score of 4.0 is the highest, followed by baseline system and InDia-w with 2.8 and 2.5 scores, respectively; the score for main InDia system is as low as 1.6. If we follow the spirit of the "long-sequence" flags from InCa (for the fully upper-cased sentence), we can hypothesize that an optimal solution for InDia would be to possibly use a single special flag for full de-diacritization of the word that would minimize its

length. Unfortunately, we will leave this modification for future work. The intrinsic statistics on the encoded texts go in line with the downstream performance described above: in the non-noised scenario, they are comparable with the baseline system, while under de-diacritization noising, the InDia alternatives that minimize the flags (`InDia-w` and `InDia-n`) show better performance than standard InDia. All scores are shown in Appendix E.

The last notable observation comes from the statistics of the tokenizer vocabulary average token length. There, we see that the CPT$_v$ score for the baseline tokenizer equals 6.83, while the tokenizer applied after InDia preprocessing has an average length of 6.91. Such a small increase (less than 0.1 characters) can be explained if we look at the diacritized subwords in the no-preprocessing tokenizer: out of 32,000 tokens, 8,454 subwords are diacritized (which comprises a quarter of the whole vocabulary and approximately a half of the Czech subwords there), but only 583 were having a non-diacritized analog. This fundamentally differs from the trends in the inline casing-optimized vocabularies described in Table 4, where up to half of the cased unique subwords have non-cased doublets. Thus, despite helping to have more consistent word splitting with respect to de-diacritization noise, the potential for increasing the lengths in the non-cased vocabularies is very restricted.

## 5. Conclusion

In this work, we introduced two inline approaches for improving tokenization stability for different noising scenarios and enhancing downstream performance. For the downstream task, we chose MT for the Czech-Ukrainian language pair. The InCa approach for inline casing shows improvement in tokenizer vocabulary elements, stability in intrinsic metrics, and on par quality with other approaches for different types of noise. It also showed improvement in MT quality for upper-cased sequences, which is explained by leveraging flags for full-sentence casing. InDia, the first to our knowledge approach for inline diacritization, also shows doubling the performance on the de-diacritized texts while showing the same performance for standard (diacritized) data; we also show that the proposed technique is stable enough to be used not only at the input side but also on the output side of translation pair. We encourage the community to use our methods for other languages and NLP tasks by publicly sharing our code in simple scripts and Python packages. While we focus on MT as the extrinsic evaluation in this paper, we hope that our methods will be useful also for other tasks where Large Language Models are being used, including e.g. multi-agent AI avatars.

## Impact Statement

This study tackles the foundational block of the NLP pipelines, namely, preprocessing of texts before applying subword tokenization. Most widely used state-of-the-art large language models are already trained through the whole pipeline, including not only tokenizers but also the model weights; thus, there is a factor of great inertia in terms of adjustments in preprocessing systems. However, we believe that small and medium-sized models, especially those aimed at specific low-resource or noisy tasks, can effortlessly leverage and benefit from our approach.

## Limitations

This research has several limitations. Firstly, we restricted the scope of languages to the single language pair of the same family, which use similar orthographical principles. Even within the European language area and Latin script-based languages, there are other orthography systems, such as German, where each noun is title-cased; thus, we cannot claim that the performance and stability of our system will be replicated for other language pairs. Similar problems stand for diacritization, as some languages use a significantly wider range of diacritics (such as Vietnamese), for which our InDia system may be inefficient.

Secondly, the tokenizers used in our comparison were based on the Unigram language model in SentencePiece (and on a similar approach in TokenMonster). Thus, it would be helpful to see how our approaches would help the NMT system if the tokenizers trained on the data would be using other principles, such as BPE Sennrich et al. (2016) or WordPiece Wu et al. (2016).

Finally, to compare the extrinsic performance of the systems, we did limited training on the MT systems. For instance, the participating systems of the latest WMT News shared task Kocmi et al. (2024) show a stable performance of several BLEU points higher than ours since they use bigger Transformer models and are trained for weeks (contrary to one day in our case). Thus, we did not claim that our algorithm reaches state-of-the-art on the Czech-Ukrainian translation pair; instead, we fixed all training parameters and compared the performance of various accessible preprocessing approaches within the same setting.

## Ethical Statement

The robustness improvement for NLU and NLG systems can be seen as a dual-use technology if an author of the text intentionally tries to prevent the automatic analysis of their texts. In many cases, such intentional noising can be used for illegal acts such as phishing or fraud. However, in countries with oppressive political regimes, the total scrapping of the

content generated by the users can be used for censorship and tracking of dissidents. Based on our knowledge, the scope of the noising scenarios examined here differs from those generally used to hide oppositional content. Still, we urge the community to bear the possibility of the robust systems they develop for evil purposes.

## Acknowledgements

This research was supported by the Czech Science Foundation project 25-16242S. and by the Technology Agency of the Czech Republic project TQ12000040 (CZDEMOS4AI). It has been using data and tools provided by the LINDAT/CLARIAH-CZ Research Infrastructure (https://lindat.cz), supported by the Ministry of Education, Youth and Sports of the Czech Republic (Project No. LM2023062).

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

## A. Diacritizations in Czech: Distances from Non-Diacritized and Most Frequently-Diacritized Bases

The InDia flags are already, by definition, longer than the single-character InCa flags. Thus, we are interested in minimization of their lengths. The first step is, of course, explicitly mentioning the flags for the characters which need to be diacritized (contrary to putting flags for each character in the sequence). But can we minimize the lengths of the flag sequences even more? A possible solution can be to leverage the logic of frequency-ordered flags, such as in the standard InCa. We can store the most frequent diacritizations of each base in the pretrained dictionary, and mark with the flags only those diacritizations that are less frequent. This is an intuitive guess, but the statistics from the training corpus may support this claim. In the dictionary creation step, we sort the diacritizations of each base by frequency. Now, for each base, we can count two values: firstly, how distant (i.e. how many additional or different diacritics) each diacritized variant is from the most frequent diacritization, secondly, how distant it is from the base (non-diacritized word). Since in the dictionary the diacritizations are ranked by frequency, we can evaluate the average diacritization distance of each rank in each of the two scenarios. We do it with the Levenshtein distance metric Levenshtein (1966). The result of this comparison is shown in Figure 2.

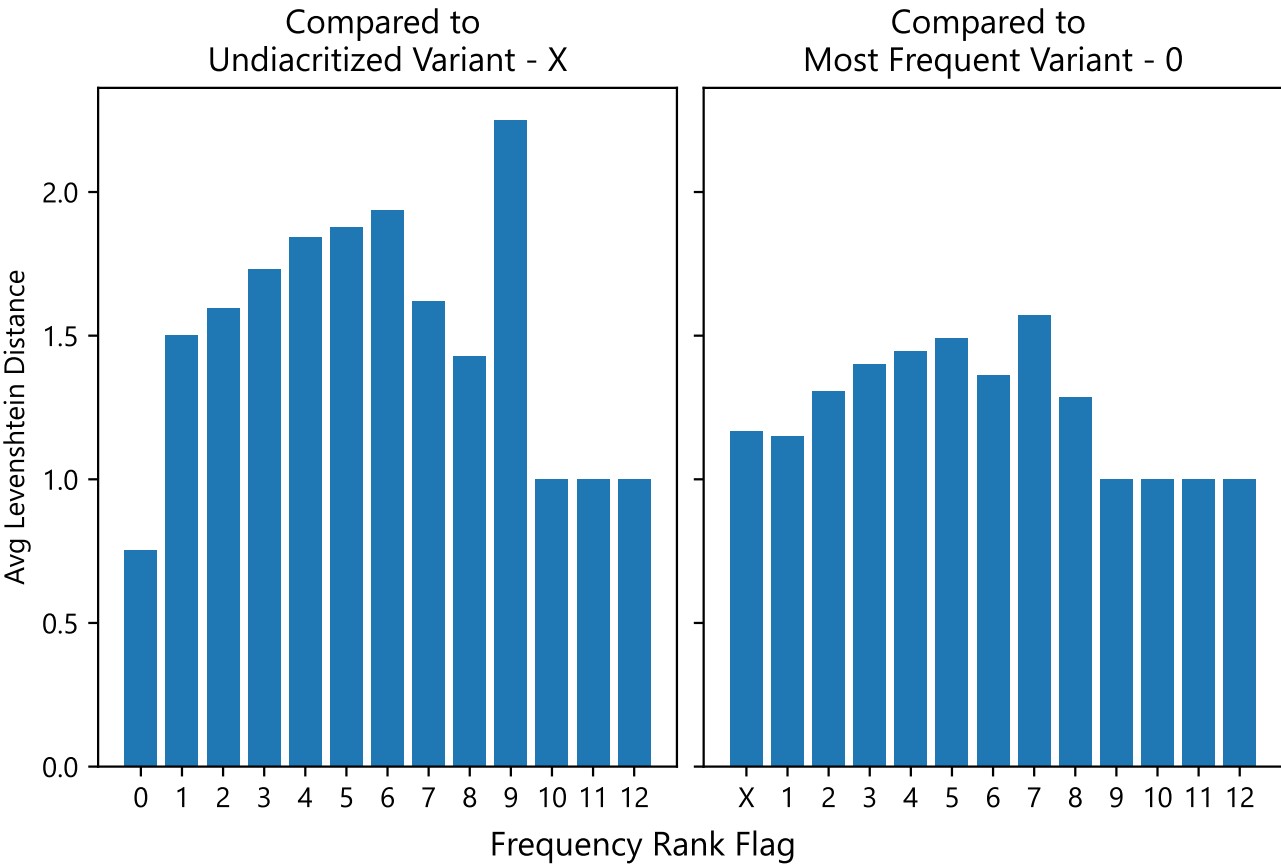

*Figure 2.* Average Levenshtein distance of the diacritization variants (ranked by frequency) for the Czech data. The x-axis represents the ranks of diacritization variants in ascending order. On the right table, we count the average distances of the diacritizations from the base (non-diacritized variant); therefore all ranks (including the most frequent, denoted by "0" flag) are shown. On the left table, we measure the distances from the most frequent diacritization (therefore "0" flag is absent); the "X" flag represents the base form in case it is different from the most frequent one. The y-axis represents the Levenshtein distance between each rank and the "starting point" (non-diacritized or most frequent diacritization), which is averaged over the whole InDia dictionary entries.

The table shows that the distribution of the ranked distances compared to the base has higher peaks and on average is approximately 1.5 characters, while if we measure the distances from the most frequent diacritization, the distribution becomes more uniform with an average of around 1.25 characters. This leads us to the suggestion that creating pretrained dictionary of the most frequent diacritizations and marking only the deviations from them would be more optimal in terms of the encoding flag length.

Interestingly, this approach resonates with the way in which diacritics are used in a number of languages, especially in consonant-based writing systems. For instance, in standard registers of Hebrew and Arabic, the vowel diacritics are not expected to be written regularly, and one is expected to predict which vowel should stay after each consonant. However, if a writer thinks that a word's vocalization would be "unexpected" in the context (usually it happens with foreign proper names or ambiguous words), one can mark a full word with diacritics. Moreover, if only one syllable is opaque and other vowels meet the expectations of a reader, one can put a vocalization diacritic only on the position "under question", which is essentially our supposed way of diacritics of only the characters "diverging" from the most common diacritization.

## B. Full Inline Casing Statistics

| Noise | Preprocessing | CPT | AR | EFF | $\text{BLEU}_{(lc)}$ | $\text{chrF}_{(lc)}$ | COMET |
|---|---|---|---|---|---|---|---|
| none | base | 3.973 | 1238 | 0.538 | $21.6_{22.1}$ | $51.3_{51.8}$ | 0.869 |
| none | inca | 3.995 | 1166 | 0.522 | $21.7_{22.4}$ | $51.4_{52.1}$ | 0.870 |
| none | inca-n | 3.592 | 1042 | 0.423 | $21.9_{22.4}$ | $51.4_{52.0}$ | 0.872 |
| none | marian | 4.033 | 1354 | 0.554 | $21.9_{22.5}$ | $51.7_{52.2}$ | 0.876 |
| none | tkm | 3.619 | 1101 | 0.492 | $21.4_{21.9}$ | $51.1_{51.6}$ | 0.870 |
| lower | base | 3.924 | 1047 | 0.539 | $18.7_{20.9}$ | $49.4_{50.6}$ | 0.849 |
| lower | inca | 3.671 | 1069 | 0.444 | $19.2_{21.8}$ | $50.1_{51.6}$ | 0.856 |
| lower | inca-n | 4.123 | 1193 | 0.527 | $19.2_{21.8}$ | $49.7_{51.3}$ | 0.855 |
| lower | marian | 4.115 | 1265 | 0.581 | $18.9_{21.5}$ | $49.8_{51.3}$ | 0.859 |
| lower | tkm | 3.760 | 1135 | 0.452 | $18.7_{20.9}$ | $49.4_{50.7}$ | 0.850 |
| $\text{rand}_{0.1}$ | base | 3.745 | 1233 | 0.549 | $19.9_{20.6}$ | $49.4_{50.3}$ | 0.839 |
| $\text{rand}_{0.1}$ | inca | 3.715 | 1085 | 0.473 | $20.5_{22.0}$ | $50.0_{51.7}$ | 0.855 |
| $\text{rand}_{0.1}$ | inca-n | 3.394 | 985 | 0.391 | $20.6_{21.8}$ | $50.2_{51.4}$ | 0.857 |
| $\text{rand}_{0.1}$ | marian | 3.907 | 1324 | 0.529 | $21.0_{21.8}$ | $50.9_{51.9}$ | 0.863 |
| $\text{rand}_{0.1}$ | tkm | 3.509 | 1063 | 0.489 | $20.2_{21.2}$ | $50.1_{51.2}$ | 0.854 |
| upper | base | 1.625 | 60 | 0.658 | $1.6_{1.9}$ | $22.5_{23.0}$ | 0.448 |
| upper | inca | 3.890 | 1134 | 0.500 | $21.3_{21.3}$ | $51.3_{51.3}$ | 0.871 |
| upper | inca-n | 3.870 | 1121 | 0.488 | $20.7_{20.7}$ | $50.7_{50.7}$ | 0.867 |
| upper | marian | 3.917 | 1213 | 0.551 | $15.5_{20.4}$ | $39.1_{50.7}$ | 0.814 |
| upper | tkm | 2.434 | 647 | 0.226 | $15.5_{17.9}$ | $46.6_{48.9}$ | 0.840 |

*Table 9.* Detailed statistics of the intrinsic and extrinsic metrics for the main inline casing algorithms, Czech-Ukrainian translation direction. The "Noise" column shows which type of noising was applied (`none` stands for standard data, `lower` for fully lower-cased, `rand`$_{0.1}$ for 10% of randomly cased words, `upper` for fully upper-cased noise). the "Preprocessing" column shows which case preprocessing algorithms were applied, where "base" means no preprocessing, "inca" means our suggested InCa system, and "inca-n" means naive InCa, "marian" shows Marian NMT suggestion by Jain et al. (2023) and "tkm" stands for TokenMonster. First three metric columns show the intrinsic metrics: "CPT" stands for character per token ratio, "AR" stands for average rank, "EFF" stands for Rényi efficiency by Zouhar et al. (2023). Three external metrics represent BLEU, chrF and COMET scores, respectively; the sub-scripted values under BLEU and chrF metrics show the lower-cased variants of BLEU and chrF scores.

| Noise | Preprocessing | CPT | AR | EFF | BLEU$_{(lc)}$ | chrF$_{(lc)}$ | COMET |
|---|---|---|---|---|---|---|---|
| none | base | 4.033 | 1189 | 0.516 | $22.7_{23.2}$ | $51.0_{51.5}$ | 0.873 |
| none | inca | 4.014 | 1113 | 0.500 | $22.7_{23.3}$ | $51.0_{51.7}$ | 0.867 |
| none | inca-n | 3.635 | 997 | 0.417 | $23.2_{23.7}$ | $51.2_{51.8}$ | 0.873 |
| none | marian | 4.197 | 1301 | 0.563 | $23.3_{23.7}$ | $51.4_{51.9}$ | 0.875 |
| none | tkm | 4.062 | 1336 | 0.503 | $22.9_{23.3}$ | $51.0_{51.5}$ | 0.870 |
| lower | base | 4.010 | 1024 | 0.519 | $19.6_{22.1}$ | $49.3_{50.6}$ | 0.847 |
| lower | inca | 3.739 | 1034 | 0.442 | $20.4_{22.9}$ | $50.1_{51.5}$ | 0.853 |
| lower | inca-n | 4.160 | 1135 | 0.504 | $19.9_{22.7}$ | $49.4_{51.1}$ | 0.854 |
| lower | marian | 4.298 | 1235 | 0.602 | $20.1_{22.8}$ | $49.6_{51.1}$ | 0.853 |
| lower | tkm | 4.237 | 1383 | 0.474 | $19.9_{22.3}$ | $49.3_{50.7}$ | 0.846 |
| rand$_{0.1}$ | base | 3.785 | 1154 | 0.527 | $21.2_{21.9}$ | $49.5_{50.5}$ | 0.844 |
| rand$_{0.1}$ | inca | 3.756 | 1041 | 0.460 | $21.5_{22.9}$ | $49.9_{51.6}$ | 0.850 |
| rand$_{0.1}$ | inca-n | 3.450 | 946 | 0.388 | $22.0_{23.1}$ | $50.3_{51.5}$ | 0.859 |
| rand$_{0.1}$ | marian | 4.069 | 1266 | 0.534 | $22.5_{23.4}$ | $50.6_{51.7}$ | 0.862 |
| rand$_{0.1}$ | tkm | 3.931 | 1289 | 0.500 | $21.9_{22.8}$ | $50.2_{51.2}$ | 0.856 |
| upper | base | 1.569 | 46 | 0.678 | $1.9_{2.5}$ | $21.8_{23.2}$ | 0.419 |
| upper | inca | 3.944 | 1091 | 0.486 | $22.8_{22.8}$ | $51.3_{51.3}$ | 0.865 |
| upper | inca-n | 3.915 | 1069 | 0.473 | $22.0_{22.0}$ | $50.8_{50.8}$ | 0.861 |
| upper | marian | 4.102 | 1181 | 0.566 | $17.6_{22.3}$ | $41.4_{51.0}$ | 0.822 |
| upper | tkm | 2.702 | 755 | 0.219 | $17.6_{19.9}$ | $47.4_{49.3}$ | 0.842 |

*Table 10.* Overview of the intrinsic and extrinsic metrics for the main Inline casing algorithms, Ukrainian-Czech translation direction. The legend is the same as in 9.

## C. Problems with Rényi Efficiency Metric

Our initial intention was to use the Rényi efficiency metric, presented by Zouhar et al. (2023). It is based on the assumption that tokenization is a noiseless transformation and is based on the concept of efficiency, which aims at penalizing the token distribution on both head and tail. The metric is theoretically based on the notion of Rényi entropy, which is a generalization of Shannon entropy. The authors show that, on a variety of tokenizers and on a set of MT language pairs, this metric correlates well with the downstream external metrics such as BLEU.

If we look at the results of casing experiments in Appendix B, we see that Rényi efficiency gives the least preference to naive InCa preprocessing; it is followed by TokenMonster, and then all other systems including the baseline without preprocessing. If we take that into context of the noising experiments (Tables 11-12), we will see the motivation behind that. The performance of the metric seems heavily dependent on the presence and frequency of the flags; and the more (and the higher rank of) the flags, the less the score of the metric. The clearest examples can be seen on the upper-case noising: no-preprocessing scenario gets the highest scores in the table, while TokenMonster obtains three times as less score (recall that it marks each upper-cased word occurrence with a token; thus it has the biggest absolute number of flags compared to any other algorithm). We understand that this should not be a fair estimate of the non-preprocessing scenario for the future work, as the quality of this system on the downstream performance was between 1.5 and 2.5 BLEU points total. Analogous trends can be seen if we compare other types of noising: for instance, InCa, being the only algorithm that uses flags in the fully lower-cased scenario (to mark the lower-cased words, for instance, in the beginning of the sentence), shows the lowest performance. This is also seen if we compare each particular system in various noising setups: for instance, naive InCa gets a lower rank of the upper-case flag in the random 10% casing scenario compared to the standard dataset, and while it is used in the lower-cased scenario without any flags, it gets its maximal score.

Can this be a problem of a particular alpha? We made the comparative graphs to see if the ranking of the systems would differ depending on the alpha value. We sampled alphas from 0 to 10 with 0.2 stride and estimated the Rényi efficiency score for each alpha. Then, we compared the performance of the systems for each noising scenario separately. The result of the evaluation on the Czech data is presented in Figure 3 (the Ukrainian data show the same patterns). Here, we firstly see that in the majority of the cases, the scores for each system decrease monotonously and do not change their ranking depending on alpha. We can also see that, while for the non-noised and randomly cased 10% scenarios the worst performance is shown

| Prepro-cessing | none | | rand$_{0.1}$ | | lower | | upper | |
|---|---|---|---|---|---|---|---|---|
| | EFF | R(f) | EFF | R(f) | EFF | R(f) | EFF | R(f) |
| base | 0.538 | - | **0.549** | - | 0.539 | - | **0.658** | - |
| inca | 0.522 | T:5 U:39 L:28 | 0.473 | T:3 U:4 L:18 | 0.444 | L:1 | 0.500 | A:3 |
| inca-n | 0.423 | T:1 U:17 | 0.391 | T:1 U:4 | 0.527 | - | 0.488 | A:3 |
| marian | **0.554** | T:0 U:1628 | 0.529 | T:0 U:3 A:2475 -A:2468 | **0.581** | - | 0.551 | T:131 U:17 A:0 -A:14 |
| tkm | 0.492 | U:35 T:3 | 0.489 | U:5 T:1 | 0.452 | - | 0.226 | U:0 |

*Table 11.* Rényi efficiency and ranks of the casing flags for various types of noising ("none" for default texts, "rand$_{0.1}$" for 10% random casing, "lower" and "upper" for fully lower- and upper-cased sentences), encoded Czech texts. The flags are denoted as follow: "T" stands for title-case, "U" – for upper-casing a word, "A" – for upper-casing the whole sentence (or a span for marian), "-A" – for ending the upper-cased span for marian, "L" – for lower-casing the word. The best (highest) scores for each column are marked bold.

by naive InCa (since it uses more tokens than the "smarter" approaches), in the upper-case scenario TokenMonster goes significantly down as it marks each word with a flag, and in the lower-cased scenario, it is InCa with the lower-case flags that lies below.[11]

The authors of the approach suggest that the increase in alpha should favor the frequent sequences to be encoded into shorter tokens. We cannot say that our evaluation supports this claim. Instead, we can say that it penalizes the systems that output numerous auxiliary tokens (which, in our case, are predominantly single-character). The only exception here is Marian inline casing that sometimes happens to even outperform the non-preprocessing scenario; this can be interpreted due to the nature of the inline casing flags that can be merged with a word, thus not creating a separate token.

In conclusion, we should say that the Rényi efficiency metric (at least in its classical version) does not favor using the characters that increase the number of separate words (and thus tokens). Thus, if we want to encode the flags separately (this is our aim – to relocate the casing information in an way of creating separate tokens), it is impossible to outperform the zero preprocessing scenario on average since any inline approach to casing would at least slightly increase the length of sentence. The case of Marian encoding shows that we can make it better if we allow the flags to merge with the words; but theoretically this does not seem a perfect solution, since if we create a digraph within a word instead of separating it from the word completely, it would not solve the problem of the possible allocation of the same words with different casings in the vocabulary.

We are aware of the theoretical criticism of the Rényi efficiency metric (for example, in Cognetta et al. 2024); however, to our knowledge, this is the first empirical evidence of the misalignment of the tokenization quality estimation and downstream performance. Therefore, we encourage the community to use this metric with caution in setups with the preprocessing techniques that require additional inline flags.

---

[11]It is less clear why TokenMonster also shows bad performance on the fully lower-cased data, as it does not use an explicit lower-case flag there. Most probably it is the result of another special token introduced by TokenMonster, "D" token that handles the deletion of the white space after this token. It is used as a way to handle the fully reversible word separation, but in an opposite logic to SentencePiece: while the latter explicitly marks the white spaces, TokenMonster by default restores white spaces between each of its tokens and then deletes them whenever the special token is used. Thus, the frequent usage of this token may skew Renyi efficiency in this case.

| Prepro-cessing | none | | rand$_{0.1}$ | | lower | | upper | |
|---|---|---|---|---|---|---|---|---|
| | EFF | R(f) | EFF | R(f) | EFF | R(f) | EFF | R(f) |
| base | 0.516 | - | 0.527 | - | 0.519 | - | **0.678** | - |
| inca | 0.500 | T:3 U:23 L:25 | 0.460 | T:3 U:4 L:19 | 0.442 | L:1 | 0.486 | A:3 |
| inca-n | 0.417 | T:1 U:16 | 0.388 | T:1 U:4 | 0.504 | - | 0.473 | A:3 |
| marian | **0.563** | T:0 U:171 -A:1617 | **0.534** | T:0 U:4 A:4012 -A:2466 | **0.602** | - | 0.566 | T:229 U:22 A:0 -A:15 |
| tkm | 0.503 | U:28 T:2 | 0.500 | - | 0.474 | - | 0.219 | U:0 |

*Table 12.* Rényi efficiency and ranks of the casing flags for various types of noising, encoded Ukrainian texts. The legend conventions follow the table on Czech data above.

## D. Example of Different Inline Diacritization Methods

In Table 13 you can find an illustration of three inline diacritization methods applied to the Czech excerpts. The first row shows the input, the next lines show the results of preprocessing and tokenization ("Baseline" means no preprocessing). InDia flags are marked in blue. For the main InDia and InDia-n systems, the flag KV marks the separator between the keys (indices of the diacritized character) and values (flags for each character diacritization), and ID is a separator if there are multiple keys. ｜°｜ means using the ring diacritization, ｜´｜ means using the acute , ｜n｜ means de-diacritizing the letter. For InDia-w, ｜N｜ means de-diacritizing the whole word, 1 and 2 mean the second- and the third- most frequent diacritizations for the same base.

| Preprocessing | Examples |
|---|---|
| input | Olympijské komisi Spojených států 
 stálá tajemnice Nobelovy komise |
| Baseline | _Olymp ijské _kom is i _Spojených _států 
 _stál á _tajemn ice _Nobelov y _komise |
| InDia | _Olymp i jske _KV 5 KV ｜n｜ _komisi _Spojenych _KV 4 KV ｜°｜ _statu 
 _KV 2 ID 4 KV ｜´｜｜´｜ _stala _ta jem nice _Nobelov y _komise |
| InDia-n | _KV 9 KV ｜´｜ _Olymp i jske _komisi _KV 6 KV ｜´｜ _Spojenych _KV 2 ID 4 KV ｜´｜｜°｜ _statu 
 _KV 2 ID 4 KV ｜´｜｜´｜ _stala _ta jem nice _Nobelov y _komise |
| InDia-w | _Olymp i jske _ ｜N｜ _komisi _Spojenych _ 1 _statu 
 2 _stala _ta jem nice _Nobelov y _komise |

*Table 13.* Illustrations for modifications of InDia preprocessing and no-preprocessing tokenization.

If we pay attention only to the splitting of bases, we can see that for all InDia variations, they are split in the same manner. Moreover, the bases are split into longer sequences, compared to the diacritized "base" text: consider the words "komisi" or "stálá", which are split into 3 and 2 tokens in "base" and are kept as single tokens in InDia systems.

We can see that both InDia and InDia-w omit flags on the word "Olympijské", since it is stored in their dictionaries. We also see that in case where the word is non-diacritized while the most frequent version of its base is diacritized, they both use flags that erase diacritization (in the case of the word "komisi", for which the most frequent diacritization is "komisí"). In case of the non-diacritized word, the words which have the diacritization different from the most frequent one tend to be over-tokenized by the no-preprocessing system, while they are kept as a whole in both InDia setups (such as the word "stálá"). Finally, we can see that if we disregard the flags, the tokenization of the bases for each word is the same in InDia

# Rényi Efficiency Comparison for Different Alphas and Different Noises, Czech Texts

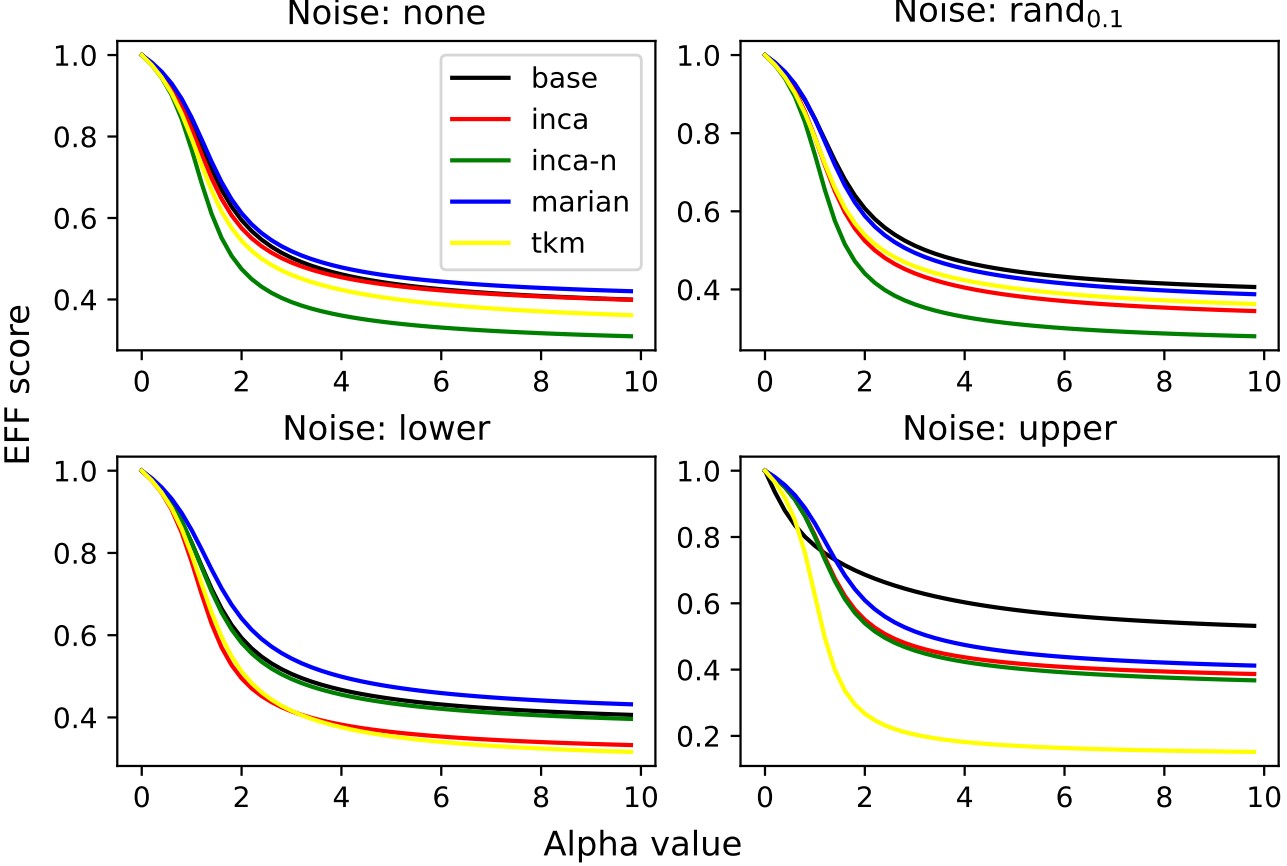

*Figure 3.* Comparison of the Rényi efficiency score depending on alpha. The subplots are created for each type of case noising, each figure shows the EFF score of each system (Y axis) with respect to alpha score (X axis).

and InDia-w.

Contrary to these two approaches, InDia-n explicitly shows every diacritization operation, disregarding frequency. Therefore, the word "Olympijské" is diacritized with the symbol "čárka", while the word "komisi" is not: despite being not as frequent word form as "komisí", it does not have any explicit diacritic and therefore is left as is. Therefore, there is no n sign in general.

## E. Intrinsic Statistics for Inline Diacritization Experiments

| Prepro-cessing | no noise | | fully de-diacritized | | 20% de-diacritized | |
|---|---|---|---|---|---|---|
| Metric | CPT | AR | CPT | AR | CPT | AR |
| base | 3.973 | 1238 | 2.829 | 418 | 3.675 | 1110 |
| InDia | 3.709 | 1112 | 1.604 | 480 | 2.922 | 878 |
| InDia-w | 3.903 | 1156 | 2.530 | 742 | 3.511 | 1040 |
| InDia-n | 1.579 | 478 | 4.092 | 1207 | 1.807 | 545 |

*Table 14.* Intrinsic metrics for different inline diacritization methods under different levels of noising. CPT stands for character per token ratio, AR stands for average rank.

Table 14 shows the intrinsic metrics for different inline diacritization methods at different levels of de-diacritization noising. The results are expected given the intrinsic performance: the highest scores are shown by baseline and InDia variations that use flags of the lowest lengths: `InDia-w` in general since it uses single-character flags, `InDia` in no noise scenario since it only marks non-frequent diacritizations, `InDia-n` in fully de-diacritized scenario since it only marks explicit diacritization. Notably, we do not see the stability under different types of noising that we could see in InCa. The main reason is suboptimality of treatment of fully de-diacritized words (or even full sequences), that ideally should use a single flag for that. We leave such improvements for further experiments.

