# OpenReview forum: "InCa and InDia: Inline Casing and Diacritization Preprocessing For Robust-to-Noise Tokenization and Interpretability"
_ICML.cc/2025/Workshop/TokShop — TokShop_

### Official Review · Reviewer_BtbH · 2025-06-09
**Accept**

**Rating:** 7
**Confidence:** 4

**Review:**

The paper proposes a preprocessing system to separate diacritics and casing from words. By representing casing and diacritics as flags prepended to the word, words can be represented losslessly while always assigning the same token ID to a family of words which only differ in casing or diacritization. The authors reason that their encoding should fix oversensitivity to casing and diacritization of language models trained on the tokenization.

The authors conduct experiments on Czech-Ukrainian MT, finding that their method substantially outperforms standard BPE under various synthetic noising settings (and slightly outperforms standard BPE in the no-noise setting).

Strengths:
- The authors design a clever and simple algorithm for preprocessing, which could be useful to the community.
- Experimental results are convincing.
- The paper (in particular Related Work) provides a nice overview of previous approaches.

Weaknesses:
- The method is quite manual and language-specific.
- I could not find the efficiency (e.g., in terms of characters per token) of the diacritization strategy (InDia). It seems from the examples in Table 12 that InDia could have highly negative effects on the length of the encoding.

Suggestions:
- I would suggest going over the paper with a Grammar Checker one more time. There are a couple of grammatical mistakes which somewhat disrupt the reading flow, for example missing articles ("also integrated into tokenization system", "at preprocessing step", "compared to baseline", ...).

---

### Decision · Program_Chairs · 2025-06-10

Accept